# Upper Ocean Response on the Passage of Tropical Cyclones in the Azores Region

Miguel M. Lima[1], Célia M. Gouveia[1,2], Ricardo M. Trigo[1,3]

*Correspondence to:* Miguel M. Lima

[1]Instituto Dom Luiz (IDL), Faculdade de Ciências, Universidade de Lisboa, 1749-016, Lisboa, Portugal

[2]Instituto Português do Mar e da Atmosfera (IPMA), I.P., 1749-077, Rua C do Aeroporto, Lisboa, Portugal

[3]Departamento de Meteorologia, Universidade Federal do Rio de Janeiro, Rio de Janeiro 21941-919, Brasil

**Abstract.** Tropical Cyclones (TCs) are extreme climate events that are known to strongly interact with the ocean through two mechanisms: dynamically through the associated intense wind stress, and thermodynamically through moist enthalpy exchanges at the ocean surface. These interactions contribute to relevant oceanic responses during and after the passage of a TC, namely the induction of a cold wake and the production of chlorophyll (Chl-a) blooms. This study aimed to understand these interactions in the Azores region, an area with relatively low cyclonic activity for the North Atlantic basin, since the area experiences much less intense events than the rest of the basin. Results for the 1998-2020 period showed that the averaged induced anomalies were on the order of +0.050 mg m$^{-3}$ for the Chl-a and -1.615 ºC for SST. Furthermore, looking at the role played by several TCs characteristics we found that the intensity of the TCs was the most important condition for the development of upper ocean responses. Additionally, it was found that bigger TCs caused greater induced anomalies in both variables, while faster ones created greater Chl-a responses, and TCs that occurred later in the season had greater TC-related anomalies. Two case studies (Ophelia, in 2017, and Nadine, in 2012) were conducted to better understand each upper ocean response. Ophelia showed to affect the SST at an earlier stage while the biggest Chl-a induced anomalies were registered at a later stage, allowing the conclusion that thermodynamic exchanges conditioned the SST more while dynamical mixing might have played a more important role in the later stage. Nadine showed the importance of the TC track geometry, revealing that the TC track observed in each event can impact a specific region for longer, and therefore greater induced anomalies.

## Introduction

Tropical Cyclones (TCs) are potentially intense atmospheric disturbances which are characterised by a low-pressure centre (eye) where strong winds curl around. Among other important properties, TCs are thermodynamic dependent phenomena, meaning that intense temperature gradients need to occur in the lower atmosphere to maintain and intensify the storm. Thus, TCs are fed from warm sea water which provide a strong moist enthalpy flux from the oceanic surface to maintain a steep temperature gradient within the lower and middle troposphere and produce massive water vapour convection (Emanuel, 2003; Holton and Hakim, 2012; Pearce, 1987).

The strong wind stress present near the surface and the associated intense curl are also shown to induce vertical mixing
and Ekman upwelling in the upper layer of the ocean. In his seminal study, Price (1981) shows, through both observed
and numerical modelling data, the evolution of sea surface temperature (SST) on the passage of a hurricane, with the
emergence of a cold wake of SST after a TC due to entrainment of water from deeper layers. This effect has since
been well studied and documented with many case studies observed, for example, the case of Hurricane Felix, in the
vicinity of Bermuda in 1995, that showed decreases in the order of 3.5-4 ºC (Dickey *et al.,* 1998), or the cases of
cyclones Nargis (2008) and Laila (2010), in the Bay of Bengal, that caused SSTs to drop by around 1.76 ºC (Maneesha
*et al.*, 2012). Additionally, several model-based works focused on either the effects caused by the TCs, or the
interaction of the TC with its own cold wake (e.g., Chen *et al.*, 2017; Zhang *et al.*, 2019).
There are also biological responses to the passage of a TC. Due to the upwelling of colder water, transport of nutrient-
rich water from the sub-superficial layer may also occur (Kawai and Wada, 2011). In this case, phytoplankton can
quickly increase in the surface layer following the rise in nutrients. This increase can be remotely sensed through
satellite observations that capture the chlorophyll-a concentration (Chl-a) increasing after the passage of a TC, since
Chl-a is generally accepted as a proxy for biological activity (Kawai and Wada, 2011; Liu *et al.*, 2009; Subrahmanyam
*et al.*, 2002; Walker *et al*., 2005).
The oceanic response, either physical or biological, to the passage of a TC depends on various aspects, most
remarkably the TC's intensity and its translation speed but also the oceanic subsurface conditions (Zheng et al., 2008).
The magnitude and significance of these aspects on the modulation of the oceanic response vary regionally, although
it is generally regarded that the most impactful phenomena are intense and slow TCs (Chacko, 2019; Price, 1981;
Price et al., 1994). Recent studies (e.g., Chacko, 2019; Pan et al., 2018; Shropshire et al., 2016) have shown that
regional differences do matter when studying the biological response. In the case of the Bay of Bengal, it was shown
that the intensity of a TC is less important, and the most meaningful aspects are the TC's translation speed and, to a
lesser degree, a pre-existing shallow mixed layer (Chacko 2019). The results from this study are important to stress
that relatively weaker TCs can also induce a strong biological response after their passage.
Until now, the Azores region has not been studied regarding its thermodynamic and biological impacts. This section
of the North Atlantic basin presents much fewer and weaker cyclones than the tropical band of the basin, with this
region being mainly a zone where TCs undergo either cyclosis or post-tropical transition into extra-tropical cyclones
or mid-latitude storms (Baatsen et al., 2015; Haarsma et al., 2013). The north-eastern Atlantic (NEA) basin, where the
Azores archipelago is located, presents significantly less TCs than the western counterpart, closer to the USA coast
(Baatsen et al., 2015; Lima et al., 2021; Haarsma et al., 2013). However, there is growing evidence of a significant
increase in the frequency of strong TCs in both western (Kossin et al., 2020) and eastern (Lima et al., 2021) halves of
the north Atlantic Ocean. The climatology of the area points to a south-north gradient in both SST and Chl-a, with a
decrease in the former and an increase in the latter (Amorim et al., 2017; Caldeira and Reis, 2017). In general, the
southern part of the Azores region offers SSTs high enough to maintain TCs, although the necessary atmospheric
conditions (e.g., high lapse rates and low wind shear) need to occur for their passage northeast through the Azores
(Lima et al., 2021). However, this area is undergoing a transition due to anthropogenic climate change and an increase
both in number and intensity of TCs is expected (Baatsen et al., 2015; Haarsma et al., 2013). Therefore, the NEA basin
is a challenging study region to assess the impact that lower intensity TCs have on the oceanic surface.
The main aim of this study is to analyse in detail the upper ocean response observed after the passage of a TC in the
Azores region, which is characterised by its lower-than-normal cyclonic activity in relation to the rest of the north
Atlantic basin. In particular, we aim to evaluate the impacts on SST and Chl-a concentration produced by important
TC characteristics (averaged maximum wind speed, average translation speed, overall impacted area, time of
occurrence, and geometry of the track). Two practical case studies, relative to Nadine (2012) and Ophelia (2017) are
then thoroughly analysed to reflect the drawn conclusions for this area.
**Data**
The main data used to evaluate the oceanic response in this study is divided into three main parts: Remotely sensed
interpolated data used to characterise the Chl-a and SST, respectively, and TC track data, which provides the necessary
additional information on the location and dynamic variables of each TC, that allow to explore the oceanic response
in the aforementioned data. Additionally, non-interpolated datasets are used for the case studies to validate the
interpolated ones; and wind-stress data is used for the Hurricane Ophelia study case.
Biological oceanic response was evaluated using a multi-sensor daily Chl-a product available through the Copernicus
Marine Environment Monitoring Service (CMEMS) in a 4 km x 4 km resolution from the end of 1997 to the present
(CMEMS, 2021b). This product, delivered by the ACRI-ST company, is based on the Copernicus-GlobColour project
and obtained by merging different sensors: SeaWiFS, MODIS, MERIS, VIIRS-SNPP&JPSS1, OLCI-S3A&S3B. The
final Chl-a product is a mix of several algorithms that consider different water conditions, such as oligotrophic,
mesotrophic, coastal, clear, and complex waters (Garnesson et al., 2019). To produce a "cloud free" product, the
resulting data was subjected to daily interpolation to fill any gaps (Krasnopolsky et al., 2016; Saulquin et al., 2019).
The lack of gaps in this dataset is particularly relevant in the context of this study since the areas analysed will be
concentrated around the TCs; it is then expected that large amounts of the analysed areas would be under cloud
coverage and, therefore, some of the analysed data is not real but interpolated values. Nonetheless, CMEMS provides
approximate uncertainty levels for this data, which we used to assess the quality of our results. For further validation
purposes we used also a non-interpolated Chl-a product generated by the Ocean Colour component of the European
Space Agency's Climate Change Initiative project (OC-CCI) (Sathyendranath et al., 2019). This dataset results from
a merge of several sensors: SeaWiFS LAC and GAC, MODIS Aqua, MERIS, VIIRS, and OLCI. ESA's OCC-CI
version 5.0 Chl-a product has 0.042º resolution and a daily temporal resolution (Sathyendranath et al., 2021).
To evaluate the physical oceanic response and to relate this to the biological one, a daily SST dataset from the CMEMS
was used, with a 0.05º resolution. This data is available from 1981 up to the near present (CMEMS, 2021a). Similarly
to the previous CMEMS interpolated Chl-a product, the SST field is also a blended gap-free analysis product, with
the present one resulting from re-processed (A)ATSR, SLSTR and AVHRR sensor data being applied to the

Operational SST and Sea Ice Analysis (OSTIA) system (Donlon et al., 2012). This reprocessed analysis product provides an estimate of the SST at 20 cm depth. The inputs to the system are SSTs at 10:30 am and 10.30 pm local time which means that the analyses roughly correspond to the daily average SST (Good et al., 2020; Lavergne et al., 2019; Merchant et al., 2013). As stated before, approximated error values for SST are also provided by CMEMS. Additionally, AVHRR Pathfinder version 5.3 collated data was used as non-interpolated data for validation. This dataset, similarly to the CMEMS one, is a collection of twice daily (averaged to daily), 4 km spatial resolution, merged SST product, provided by NOAA's National Centers for Environmental Information (Saha et al., 2018). The merge of this data, however, is only used to spatially collate the data, as it is a single instrument measurement (AVHRR) onboard NOAA-7 through NOAA-19 Polar Operational Environmental Satellites (POES).

Wind stress data to assist in the analysis of the Hurricane Ophelia study case was provided by NOAA's CoastWatch dataset available at https://coastwatch.pfeg.noaa.gov/erddap/griddap/erdQMstress1day_LonPM180.html. This dataset is derived from wind measurements obtained from the Advanced Scatterometer (ASCAT) instrument on board EUMETSAT's MetOp satellites (A and B) at a daily 0.25º resolution, from 2013 to the present. ASCAT presents a near all-weather capacity (not affected by clouds), as it operates a frequency in C-band (5.255 GHz), therefore, minimizing the number of missing values in predominately clouded areas such as the case of TC paths.

The TC track data is made available by the *International Best Track Archive for Climate Stewardship Project* version 4 (IBTrACS v4) free access dataset (Knapp *et al.*, 2009). This dataset contains global information regarding TC activity since the 1851 hurricane season up to 2020. It aggregates variables such as TC geographical location, maximum wind speed, minimum sea level pressure, and storm radius estimation based on wind intensity, measured at 6-hour intervals (original dataset interpolates for increased resolution, at 3-hour rates, however this interpolation only includes the geographical location). For the 1998-2020 period, the Azores region experienced the passage of 62 individual TCs accounting to 642 6-hour observations that are categorised in the following intensities according to the Saffir-Simpson hurricane wind scale (Taylor *et al.*, 2010):

- 148 tropical depression observations.
- 389 tropical storm observations.
- 85 category 1 hurricane observations.
- 18 category 2 hurricane observations.
- 2 category 3 hurricane observations.

The full TC tracks can be better visualised in Fig. 1, with the left panel showing the full track for all these 62 TCs observed in the NA basin for the 1998-2020 period and the right panel showing a zoomed view relative to the considered Azores region. Tropical depression observations (dark blue in Fig. 1, right panel) account for 23 % of the total observations and will not be considered in this study, as they present the lower branch of intensities with winds below the 34-kt (18 m/s) threshold. Therefore, a total of 494 TC 6-hour observations were considered for this study.

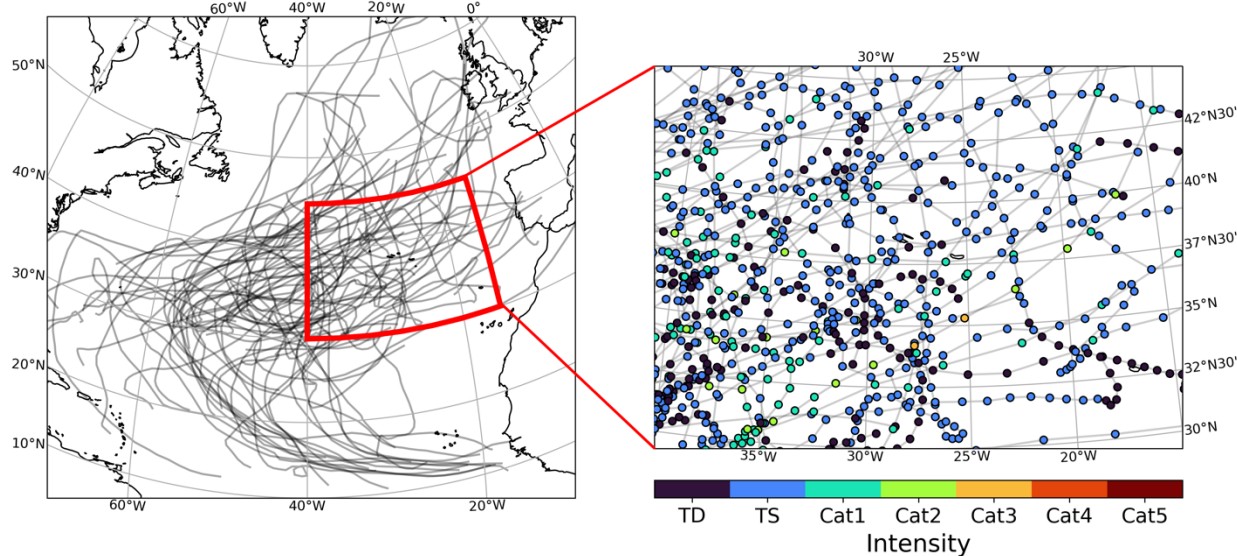


**Figure 1 - Left panel: North Atlantic basin and the tracks of all the TCs that went through or occurred inside the study**
**region (shown by the red outline). Right panel: Zoom of the previous red outline, with each TC observation marked in**
**different colours for intensity (TD: Tropical Depression; TS: Tropical Storm; Cat1 - Cat5: Hurricane category according**
**to the hurricane Saffir-Simpson wind scale).**
Since the interpolated datasets used for most of this study do not share the same time frame and to better encapsulate
full years of data, the timeframe of the present study will be from January 1st of 1998 to December 31st of 2020.
Moreover, while we have extracted all the data described above covering the entire North Atlantic basin, we will focus
on the area around the Azores archipelago, delimited by the 15º W and 40º W meridians and between the 30º N and
the 45º parallels (Fig. 1).
**Methodology**
The region of study was chosen due to its nature regarding TCs, since it is an area with fewer and less intense tropical
storms (Hart and Evans, 2001; Lima et al., 2021; Ramsay, 2017). Generally, tropical cyclosis and post-tropical
transition occur here (Baatsen et al., 2015; Haarsma et al., 2013). Because of these aspects, it corresponds to a much
less studied area and is a good region to characterise oceanic biophysical effects after the passage of (generally) weaker
TCs at higher-than-tropical latitudes and to compare the obtained results with previous literature.
To cope with large amounts of data, the bio-physical response was evaluated within a small area around individual
locations obtained for each TCs' best-track location. For this, we used the approximated quadrant radius given by the
IBTrACS v4 dataset. This dataset provides different types of radii depending on the considered isotach, for this study
we used the 34-kt isotach as it corresponds to the lower-bound for the Tropical Storm status according to the Saffir-
Simpson hurricane wind scale (Taylor et al., 2010). Since the considered area of analysis falls above the 34-kt isotach,
tropical depressions were not considered (exact partition of intensities is given at the beginning of the *Results* section).
There are some missing radii values in the middle of TC tracks and, to correct those, a simple linear regression was

applied. To illustrate the application of this methodology we present the study cases in the *results and discussion* section, for hurricanes Ophelia (2017) and Nadine (2012). From inside this area of analysis, we may retrieve the Chl-a concentration and SST at their respective resolution. The analysis inside the considered area was performed using histograms, in which each pixel inside the 34-kt isotach contributes to that TCs histogram.

To analyse the TCs' impact on their passage, inspiration was taken from Kawai and Wada (2011), who computed the climatic monthly standard deviation of Chl-a on 0.25º grids over a 5-year study period. Here, we compute the daily normalized anomaly from the climatological value (in standard deviation units). For this, we first calculated the climatological mean and associated standard deviation of both Chl-a and SST values for the region that is impacted by each TC on the day of analysis. This is achieved considering the 3 days before and 3 days after the day of analysis, totalling one week that is then retrieved from the entire study period of 22 years, thus ensuring a larger sample and a smoother continuous curve. Then, we compute the mean value in the same area (in which only the TC area was considered) for the day of analysis, and finally, we calculate the normalized anomaly from the climatology on that day. This analysis was performed considering 30 days before and after each TC to allow then the analysis and identification of an ideal window to compute the induced anomalies. To compute this ideal window, we searched for the maximum difference between the number of standard deviations over the climatological value before and after the storm.

To compromise between having the maximum difference and ensuring a time window as close as possible to the storm (to minimize external factors to the TC), we performed a sensibility study on the length and location of the considered time window. First, we analyse the overall maximum difference in the 61-day period (including the day of the storm) and then search for a secondary maximum value that is within 10 % of it considering a smaller sample of days, decreasing in groups of 5 days each time this search is made (e.g., the first iteration would be 25 days before and 30 after, the second 30 before and 25 after, the third 25 before and after, etc.), until an optimum maximum difference value is identified. With this window defined, the induced (or TC-related) anomalies are simply the difference between the daily values of Chl-a or SST after and before the TC.

As an example of this methodology, Fig. 2 shows the Chl-a standard deviation over the climatological value in the case of Hurricane Nadine. In this case, only 15 days around the TC are shown for clarity. We can see that the maximum difference is obtained between 8 days before and 1 day after the storm (ΔChl-a max). However, when we take into account the compromise of considering windows located as close as possible to the occurrence of the TC over the region, we see that the value found between 4 days before and 1 day after is within 10 % of the absolute maximum. This methodology is then applied to all 6-hour observations individually and for each TC, thus resulting in two groups of induced anomalies (per TC and per 6-hour observations) where we can study these with respect to the TCs averaged (per TC) or instantaneous (6-observations) characteristics.

To address the possibility that some pixels are overlaid on top of each other, which would contaminate the analysis, as observed in the case of the slow erratic Hurricane Nadine (presented in the *results and discussion* section as a study case), we did not take into consideration the days in which the TC is over the aforementioned overlaid region. In the

case of these pixels, the day considered to be after the TC is the day after it has completely passed over the area (i.e., the observations in that pixel during the days the TC is still over the area are discarded). However, when we consider independent 6-hour observations, this caveat cannot be accounted for since we have no way of knowing if that area has been influenced or not by the TC before, for how long, or even if a future observation will impact the area.

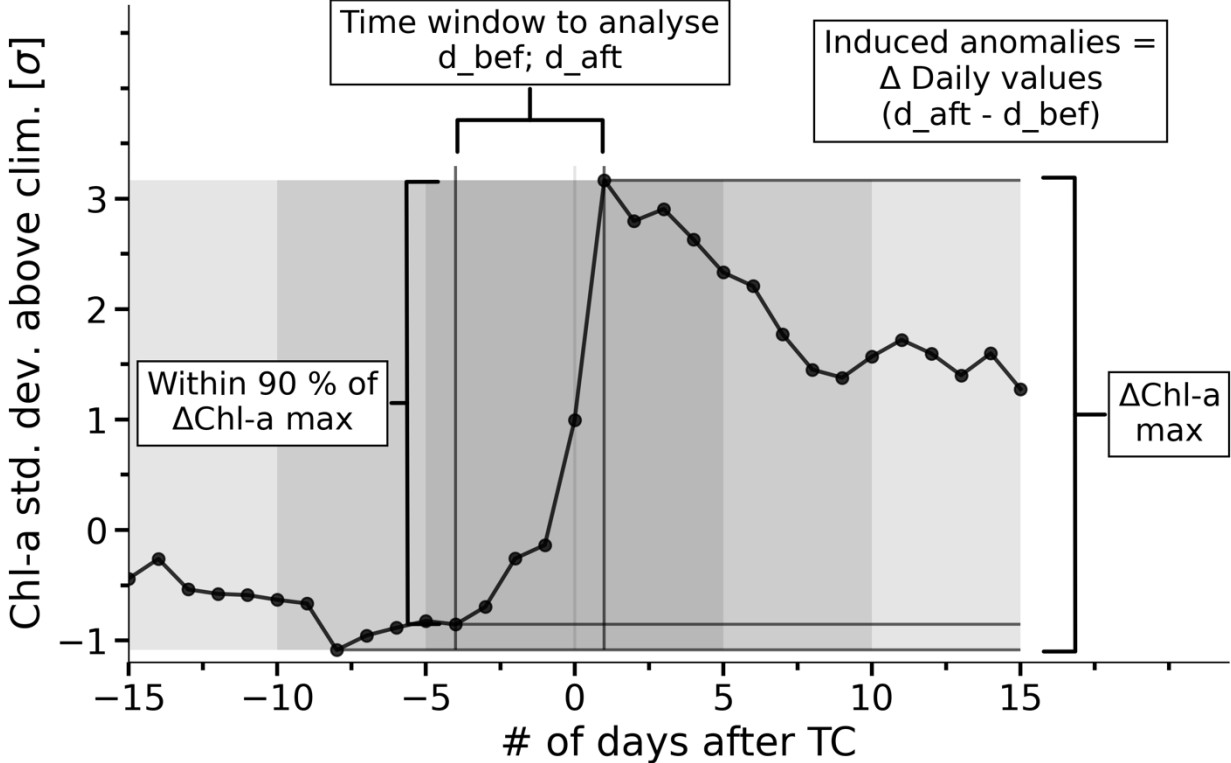

**Figure 2 - Schematic of the applied methodology for each TC. Black line shows the number of standard deviations from the climatological values for the area surrounding Hurricane Nadine. A detailed description of this methodology can be found in the text.**

As previously mentioned in the *Data* section, the interpolated data used for this study is expected to encounter some regions where clouds are to be expected due to the presence of the TCs. To account for this potential caveat, we looked at the uncertainties associated with the data before and after the TCs, as well as during the TC (e.g., day 0 in Fig. 2), to evaluate if there were clear increases in uncertainty for cloud covered situations.

Two case studies were looked at in greater detail: Hurricane Ophelia (2017) and Hurricane Nadine (2012). The former was performed to assess the different impacts along the lifecycle of the storm, and different histograms were produced for smaller portions of the TC. The latter was made to analyse the possible increasing impacts the storm geometry could cause. Additionally, these study cases were used as validation for the interpolated "cloud-free" data, where a comparison was made between the non-interpolated and the interpolated "cloud-free" data described in the *Data* section.

**Results and Discussion**

Applying the mentioned methodology leaves us with a large pool of induced anomalies, from which we can now evaluate the distribution of these TC-related anomalies for both the Chl-a and SST as shown in Figs. 3a and 3b in the form of histograms of induced Chl-a and SST induced anomalies, respectively. Both variables present a large impact after the passage of TCs, with the Chl-a presenting a mean response of positive 0.050 mg m$^{-3}$ and the SST showing a mean response of -1.615 ºC. Figs. 3c-f show the corresponding distributions as a function of the cyclone's intensities (Figs. 3c and 3d) and translation speeds (Figs. 3e and 3f). To make these distinctions, we chose only the high values (either regarding intensity or translation speed) to be those above the third quartile and the lower values to be those below the second quartile.

Firstly, regarding intensity (Figs. 3c and 3d), we have the induced response of the most powerful intensities in orange and the weaker ones in blue. Regarding the impact as a function of intensity it is possible to observe that more powerful TCs tend to induce a stronger biological response than weaker ones, which have a mean response closer to zero. It is also important to note that the more powerful TCs have a response that is much more skewed towards extreme positive values of Chl-a. Fig. 3d also shows a great impact regarding different intensities in SST, in which even weaker TCs show a substantial mean response of -1.517 ºC and nearly all the analysed pixels showing negative induced anomalies. Important to note the nearly bimodal nature of this distribution, which can be attributed to both the earlier phase of TCs (more energy being drawn from the ocean) resulting in more negative SST values, and the less negative corresponding to the later part of TCs since baroclinic instabilities are more prevalent than the action of moist enthalpy flux from the ocean at this phase (Baatsen et al., 2015; Emanuel, 2003). Powerful TCs induced a more varied distribution of induced anomalies, with a mean response of -1.694 ºC.

Regarding the different translation speeds, Fig. 3e shows that, for biological responses, faster TCs show a greater mean value of +0.060 mg m$^{-3}$. This difference is not as remarkable as the one in Fig. 3c. On the other hand, the SST response (Fig. 3f) seems to be weakly impacted by the TC's translation speed, with slower TCs having a slightly stronger impact than faster ones, while the mean response values do not differ as much as the ones in Fig. 3d. Additionally, even if faster TCs do not affect the SST response as much as slower ones, the mean value is still close to what is seen in the general case in Fig. 3b, and most of the impact is towards negative SSTs.

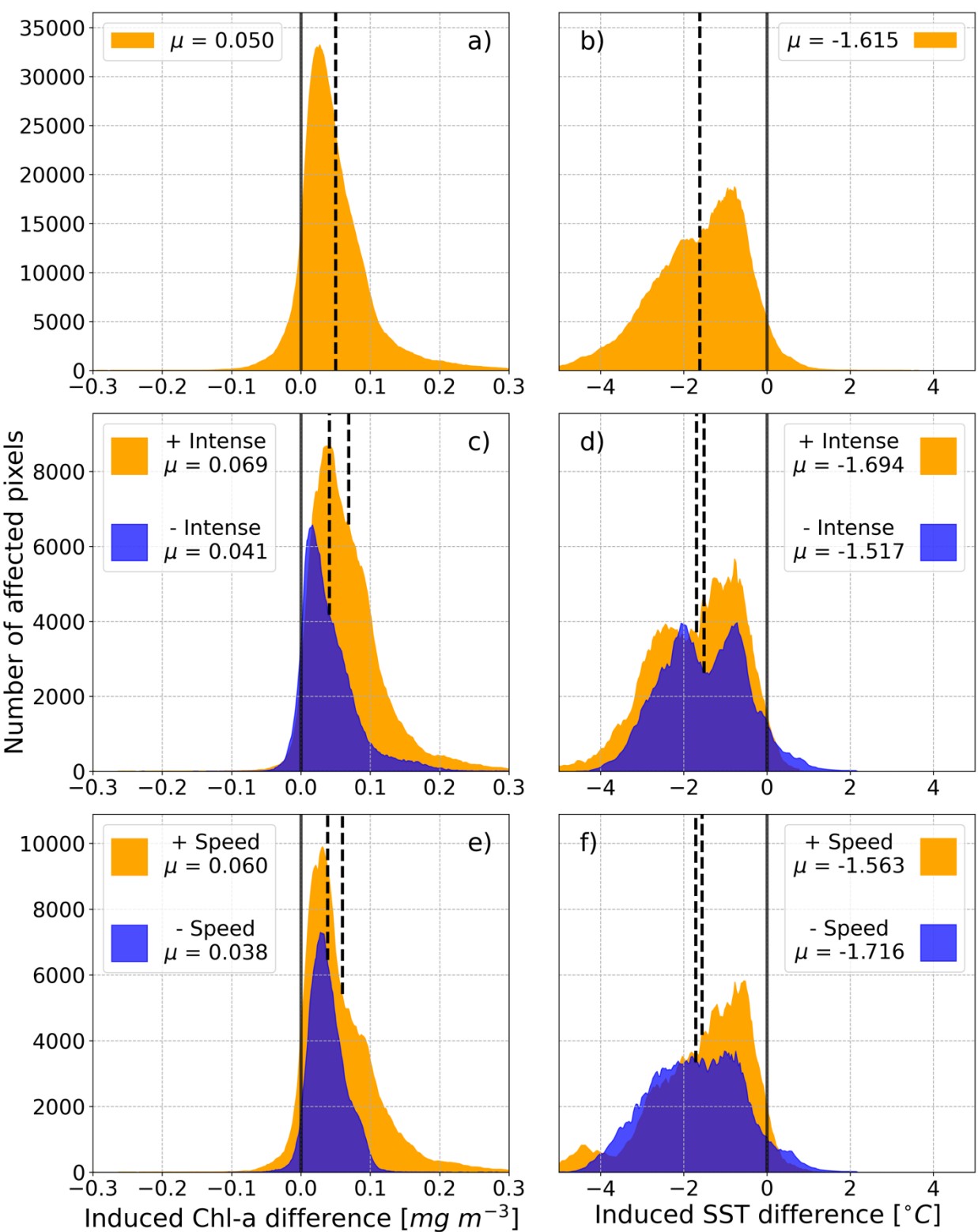

235

Figure 3 - Histograms for the: a) Total Chl-a and b) SST induced anomalies; c) Chl-a and d) SST induced anomalies after weak (blue) and powerful TCs (orange); e) Chl-a and f) SST induced anomalies after slow TCs (blue) and fast TCs (orange).

**Each subplot histogram presents the respective population mean value (μ) in a dashed black line, and the zero value in a**

**grey line.**

**240**     To quantify these relations, Fig. 4 shows the storm-averaged induced anomalies compared to the averaged maximum

**241**     wind and average translation speed. The linear regression is also shown for each of the comparisons, with nearly all

**242**     results significant at the 95 % statistical level. According to these plots, only the translation speed in relation to the

**243**     SST induced anomalies (Figs. 4d) did not show a significant relation at the 95 % statistical confidence level (marked

**244**     by the dashed regression line). Regarding the mean wind (Figs. 4a and 4c), and therefore the TC's average intensity

**245**     within the Azores region, the linear regression showed significant values, upwards of 0.5 for Chl-a and -0.3 for SST

**246**     induced anomalies. In the case of Chl-a, like observed in Fig. 3, the relation is positive while with SST this relation is

**247**     negative. Considering the translation speed, the relation is equally positive and significant for biological responses (r

**248**     = 0.416).

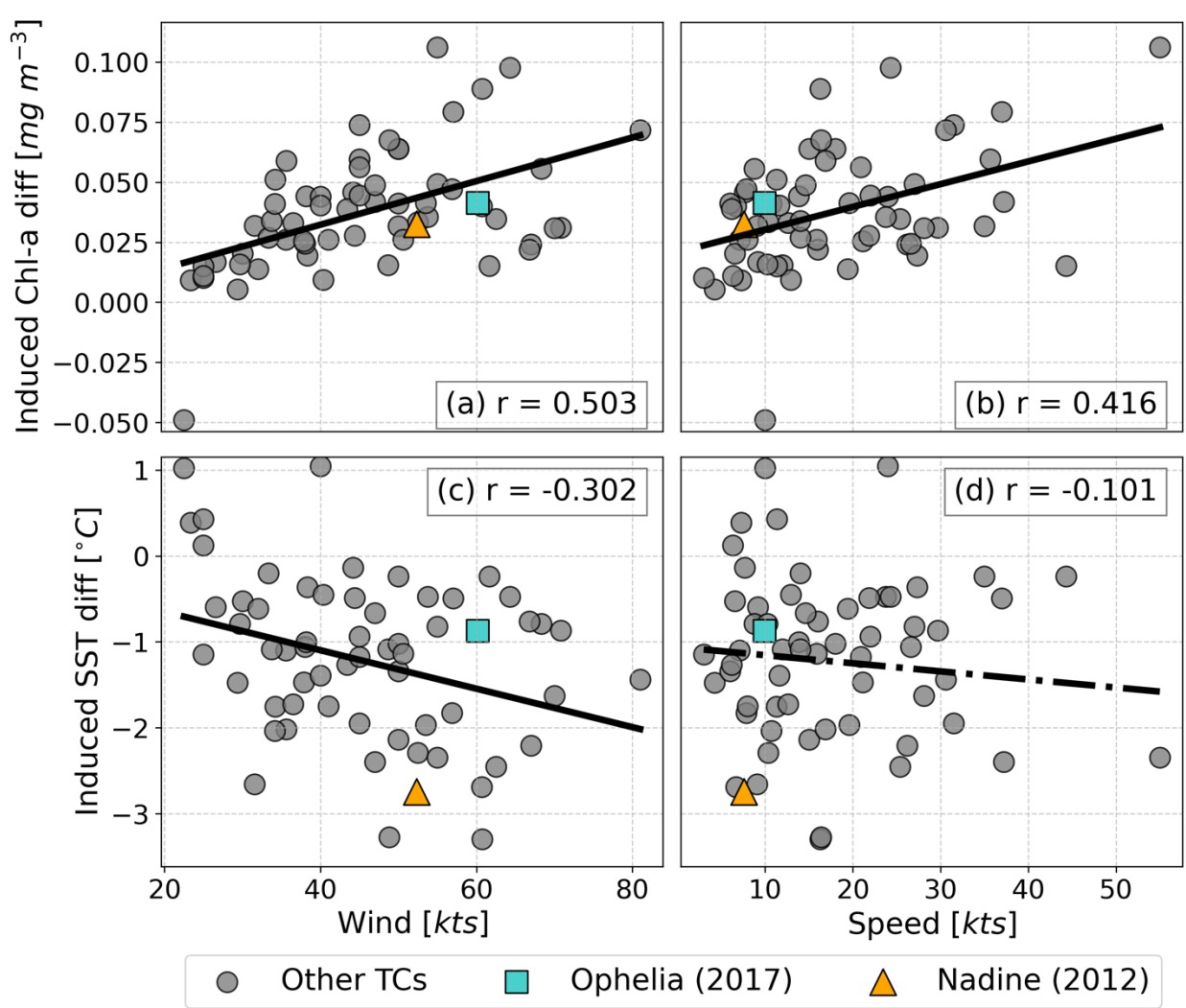

**249**

**250**     **Figure 4 - Linear regression of Chl-a (top panel) and SST (bottom panel) induced anomalies for each TC, respectively, when**

**251**     **compared with average winds in knots (left column); and average TC translation speed in knots (right column). In each**

**plot the Pearson R is presented, and the regression's significance is marked by the type of line used in the regression, with**
**a dashed line representing non-significant at a 95 % confidence level, and a solid line representing a regression significant**
**at the 95 % confidence level.**
Further analysis of other TC characteristics requires a different approach. Fig. 5 shows similar relations to Fig. 4 but
considering 6-hour observations instead of total TC mean values. This is made to account for the possible error that
averaging a whole TC may create since the cyclone's characteristics may change substantially along its lifetime. This
analysis, however, does not consider the possibility of superposition in pixels from observation to observation – i.e.,
from a TC that either moves slowly or whose track is more erratic, ending up covering the same area for several
hours/days. This caveat was not present in Fig. 4 since we considered the TC lifetime as a whole and could then
disregard the days of superposition. Using 6-hour observations, we can study several types of characteristics that
change between observations, such as the impact area or the time of season when it occurred, adding to the already
seen maximum wind speed and translation speed.

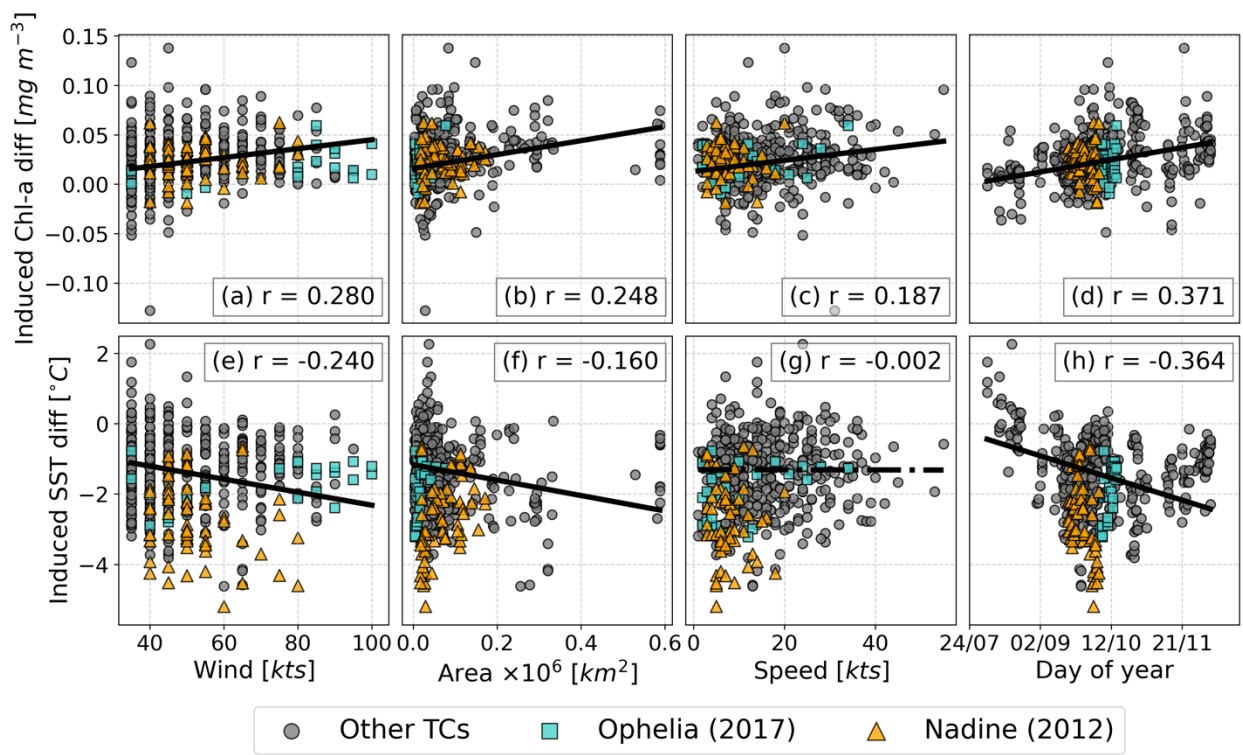


**Figure 5 – Same as in Fig. 5 but considering individual 6-hour observations. Two columns are added: (b) and (f) with respect**
**to the area affected by that observation; and (d) and (h) with respect to the time of the year when that observation occurred.**
Considering then the maximum wind speed per observation (Fig. 5a and 5e), both variables are significantly related
to this characteristic, which is expected considering the analysis made in Figs. 3 and 4. As previously noted in the
form of histograms in Fig. 3, most observations show a positive impact regarding Chl-a and, especially for SST as
most fall below zero, a negative change after a TC. The affected area (Figs. 5b and 5f) also presents a significant
relation, although less intense than that observed with the maximum winds. However, it should be noted that this
variable is linked to the mean winds, since more intense cyclones tend to be larger than less powerful ones, but also
to the storm phase, since storms nearing their post-tropical transition tend to grow larger (Knaff et al., 2014).
Translation speed is the less correlated variable from those studied (Fig. 5c and 5g), with only the biological response
seeing a positive relation to this characteristic, agreeing with the previous results from Figs. 3 and 4. The time period
in the season in which the TC occurs seems to also be important for the magnitude of the average induced anomaly
seen in both variables (Figs. 5d and 5h) with late occurrences in the season showing greater responses respective to
the signal of induced anomalies seen in Figs. 3a and 3b. Lastly, a geographical correlation was concluded not to be
relevant for this study (not shown), as both variables were correlated with both latitude and longitude, and only non-
significant relations were found.
The results presented so far in this study result from interpolated "cloud-free" data and should be quality assured to
guarantee the integrity of the conclusions made previously. As mentioned in the *Data* section, CMEMS provides
measures of uncertainty for the used Chl-a and SST datasets. Thus, we have explored these values at different periods
as a first step in validating the quality of the data. Figure S1 shows the associated uncertainty with respect to the
absolute observed values both for Chl-a (top panels) and SST (bottom panels) for three different periods surrounding
a TC event (before, during, and after), and a randomly drawn sample of the same size as the data analysed in the other
subplots. It becomes immediately clear from these plots the considerably different magnitude of uncertainty for this
data, with Chl-a (Figs. S1a-d) ranging from 25 % to 45 % considering all moments, while SST (Fig. S1e-h) does not
commonly surpass 0.4 % with a mean error around the 0.25 %. The randomly drawn sample of data gives a rough
idea of the average uncertainty we can find in this dataset, with Chl-a (Fig. S1a) presenting values around 35 % and
SST (Fig. S1e) around 0.25 %. Additionally, we should consider three distinct moments of analysis, namely before
and after the TC passage, which corresponds to the data used to compute the induced anomalies, and during the TCs,
which should be the moment with most cloud-cover over the studied regions. Looking first at Chl-a (Figs. S1b-d) we
see the progression from near normal uncertainty before the TC (Fig. S1b) to an increase during TCs (Figs. S1c),
likely due to the larger cloud-covered area in that situation. After the storm (Fig. S1d) however, the uncertainty
substantially decreases reaching values below the randomly drawn sample (around 30 % compared to 35 %). For the
SST (Figs. S1f-h), the associated uncertainty does not fluctuate substantially, constantly being below the 0.3 % mark.
Additionally, the variation that has been identified before, with Chl-a increasing and the SST decreasing, is noticeable
in both variables.
Visible in Figs. 4 and 5 are two case studies: Hurricane Ophelia in 2017 (squares) and Hurricane Nadine in 2012
(triangles). These case studies were chosen based on the presented characteristics, coupled with the amount of
sampling data within the region. Hurricane Ophelia (2017) was chosen due to its large intensity in the region (squares,
Fig. 4 and 5), reaching a category 3 intensity in the Saffir-Simpson hurricane wind scale, something abnormal for the
region (Lima *et al.*, 2021). The complete TC track can be seen in Fig. 6a insert. Besides the large intensity, Ophelia's
genesis took place inside our study region which enabled us to study different phases of the storm and its impacts on
the ocean surface in the region. Even though hurricane Ophelia was so intense, this storm impacted a very small area
(Figs. 5b and 5f) particularly when compared with the other case study, Hurricane Nadine (2012). Hurricane Nadine
(Fig. 7a) was chosen due to its large sampling, relatively high intensity (maximum category 1) and great impact area
(second highest in this study, considering cumulative area of impact). The large impacted area was amplified by the
geometry of the storm's track (i.e., many overlaid observations). Only the final stage of Hurricane Nadine was caught
within the study region, producing an ideal case study to analyse the impact of a less intense storm that heavily
impacted a particular region due to its geometry.

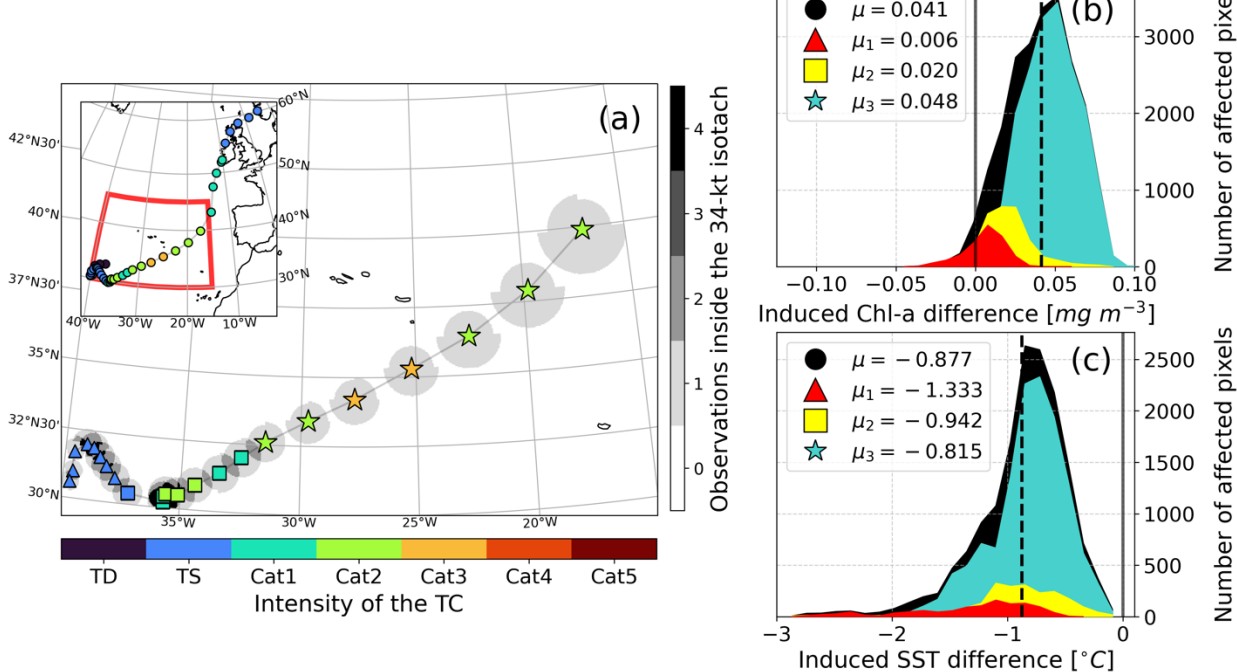


**Figure 6 - Case study for Hurricane Ophelia, in 2017, with its track on the left panel (scatter marker colour scheme**
**represents intensity as in Fig. 1), as well as the affected area around the cyclone (marked as the 34-kt isotach) with shading**
**according to the number of pixels overlapping. Inside, there is an inset with the full track and the region of study marked**
**with a red box. Ophelia track is divided in three phases: Histograms show induced Chl-a (b) and SST induced anomalies**
**(c), by phase of the storm (colours) and in total (black). The phase of the storm is marked in (a) as triangles (genesis),**
**squares (maturing), and stars (mature) and correspond to the colours in (b) and (c).**

For the case study of Hurricane Ophelia (2017), three different phases of the storm were studied, corresponding
approximately to: cyclogenesis (Fig. 6a, triangles), maturing (Fig. 6a, squares), and mature hurricane (Fig. 6a, stars).
There are 23 total observations; the first two phases encompass 8 observations and the last one 7. Each of these phases
has its own histogram in Figs. 6b and 6c (shown in colours), for the induced Chl-a and SST TC-related anomalies,
respectively. The histograms are inserted in a larger one (in black), representing the total induced anomalies caused
by Ophelia and therefore, the sum of all three phases will result in the bigger histogram. Regarding the Chl-a induced
anomalies (Fig. 6b), Ophelia seemed to have a higher impact towards the end of its track in the region of study, when
the storm had the highest intensity and the mean values of the induced anomalies increased along the track. Even at
the storm's genesis, the induced anomalies were mostly positive with a mean value of +0.006 mg m$^{-3}$ reaching +0.048
mg m$^{-3}$ in the most intense phase. In contrast, the SST induced anomalies (Fig. 6c) present the highest mean response
(-1.333 ºC) at the initial phase. The SST induced anomaly is then seen decreasing as the storm goes on, with the last
phase weighing the most in the general distribution (as was seen for the Chl-a). The highest SST impact of the storm
during the initial phases may reflect that this is the phase of the storm with highest interaction with the ocean, regarding
thermodynamic exchanges (Emanuel, 2003).
As a further insight to Ophelia's interaction with the ocean surface, Fig. S2 shows the mean modulus of wind stress
on the surface, by day of analysis (Fig. S2a) and by Ophelia's 6-hour observations (Fig. S2b). Marked in both these
plots are the analysed periods in corresponding colours and marker type to Fig. 6. These plots exceed the original
study region, in order to fully encompass the TCs entire lifetime. There is a significant relation between the increased
mean modulus of the wind stress and the evolution of the TC in time. This increase may be related to the increase in
the storm's intensity. As Ophelia reaches its maximum intensity, so does the observed interaction with the ocean,
decreasing afterwards as the storm moves north-eastward and undergoes post-tropical transition. This observed
interaction with the ocean might be the reason for the maximum induced anomaly of Chl-a being observed at the end
of Ophelia's passage over the study region, inducing the mixing of the superficial layer.

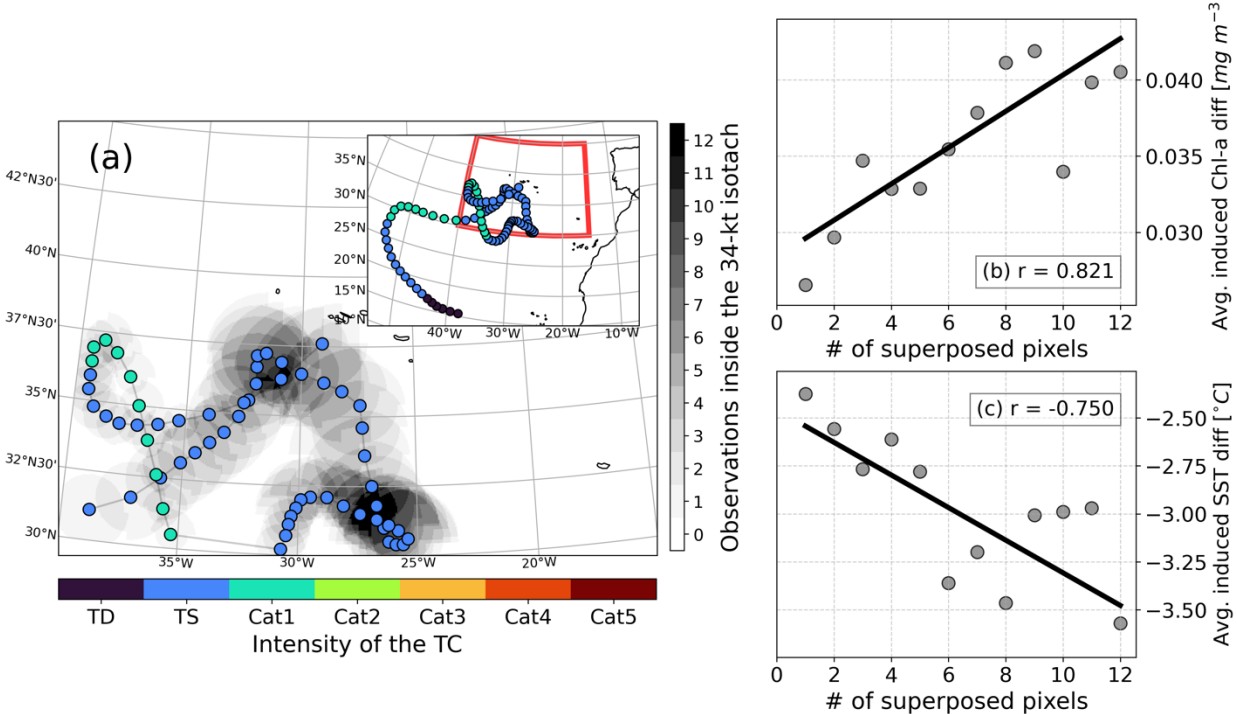


**Figure 7 - Case study for Hurricane Nadine, in 2012, with the left panel the same as in Fig. 6. For Nadine, plots (b) and (c)**
**pertain to the average induced Chl-a and SST induced anomalies, respectively, based on the amount of superposition**
**verified in each pixel.**
Hurricane Nadine's (2012) case study shows very different behaviour and impact during its lifetime to that of
Hurricane Ophelia. In this case, we present scatter plots of the averaged induced anomalies for the areas (Figs. 7b and
7c) corresponding to the superposition of pixels, i.e., the number of repeated observations inside the 34-kt isotach due
to storms track geometry (as seen in Fig. 7a). The conclusions drawn regarding the Chl-a and SST induced anomalies
are similar and significant in this case study: The more time the TC spent over a certain area the more this area became
affected by its passage, with large TC-related anomalies registered in both variables compared to less superposed ones
(over 0.040 mg m$^{-3}$ and -3.500 ºC for Chl-a and SST, respectively at 12 superposed pixels), and all cases being positive
(negative), for Chl-a (SST). It is possible to hypothesise that the translation speed also had a relevant role in these
results, with a slower TC (Nadine was one of the slowest TCs in this study, as seen by the closer observations in Fig.
7a and by Figs. 4 and 5) spending more time over a region and therefore producing larger induced anomalies.

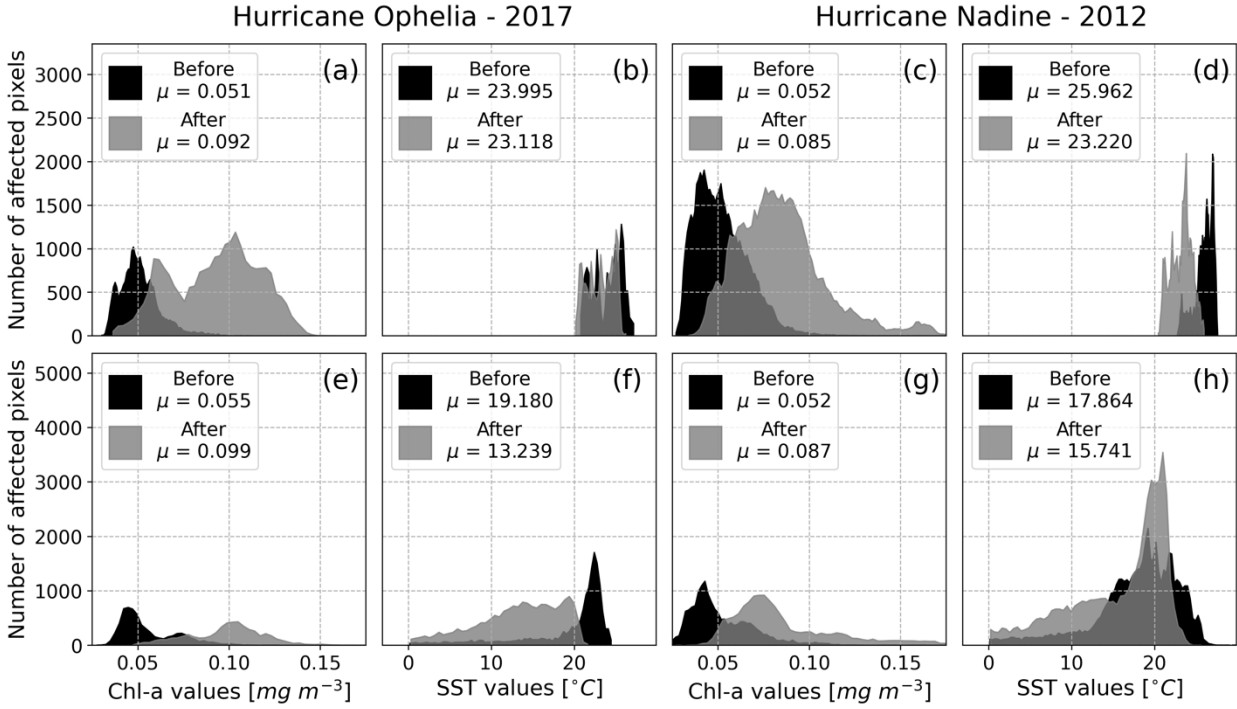


**Figure 8 – Comparison between interpolated "cloud-free" data (top row), and non-interpolated data (bottom row), for**
**Hurricanes Ophelia (2017) and Nadine (2012). Values for non-interpolated data were obtained with the same methodology**
**as the ones presented before and represent the exact same days of analysis. Mean values for each histogram are presented,**
**with black histograms representing the situation before the TC and the grey ones the situation after.**

For these two case studies, we considered an additional quality assessment exercise, by comparing the interpolated
"cloud-free" data to similar non-interpolated datasets. Figure 8 shows the histograms obtained for Ophelia and Nadine
for the situations before and after the TC, independently, since non-interpolated data cannot be correctly subtracted as
corresponding pixels may not be available. Overall, and despite the different number of observations considered, the
Chl-a presents the same average response between the different types of data for both TCs, with non-interpolated data
having an observed mean increase of 0.044 mg m$^{-3}$ for Ophelia (Fig. 8e) compared to 0.041 mg m$^{-3}$ for interpolated
data (Fig. 8a), with these values representing the difference in the mean values shown in Fig. 8. Likewise, non-
interpolated data reveals an increase of 0.035 mg m$^{-3}$ for Nadine (Fig. 8g), compared to 0.033 mg m$^{-3}$ for interpolated
data (Fig. 8c). Looking at the histograms, the shape of the data itself does not differ much between the different types,
with peaks more or less located over the same values and distributions ranging the same values. However, for the SST
variable, despite both TCs presenting relatively similar decreases between both types of data, the non-interpolated
data has a wider range of values, and the peaks do not correspond so closely. This, however, may be due to the process
of data collation. In this process, some pixels are averaged with incorrect ones, resulting in unrealistic values in some
areas. This can be identified by the unrealistic SST seen in Figs. 8f and 8h, with values that do not support TC
development around 18-19ºC and, so far as reaching 0ºC. Nonetheless, interpolated SST data does show very low
uncertainty as verified before (Fig. S1).
**Final remarks**
The current study provides the first general assessment of the bio-physical oceanic response to the passage of TCs in
a relatively low cyclonic activity area such as the region near the Azores archipelago. It is important to stress the
efficiency of identifying the precise timing and associated spatial impacts of all TCs using remotely sensed products
that rely on interpolated areas to fill existing gaps due to cloud coverage or lack of satellite imagery.
Over the Azores region, the existence of a bio-physical response after the passage of a TC was identified from the
analysis of Chl-a and SST datasets, which produced signatures of positive Chl-a and negative SST induced anomalies.
This signature is more intense for the SST analysis, in which the passage of a TC results in nearly all observed pixels
to have a negative (i.e., cooling) induced anomaly. On average, TCs produced positive induced anomalies in the order
of 0.050 mg m$^{-3}$ regarding Chl-a and a mean SST cooling of 1.615 ºC.
The more powerful TCs tend to produce more intense bio-physical oceanic responses, which agree with previous
literature on the topic (Chacko, 2019; Price, 1981; Price et al., 1994). TC translation speed was also found to be
associated with the induced anomalies, although the relationship was found to be positive and significant in the case
of Chl-a while it was not significant at the 95 % statistical confidence level for SST. The impacted area was also found
to be significantly linked to the oceanic response. However, the sensitivity to the impacted area can rise due to several
other factors: slower TCs impact larger areas (due to track geometry); more intense TCs impact larger areas (Knaff et
al., 2014); and TCs nearing post-tropical transition are generally larger (Knaff et al., 2014). These effects, either
individually or combined, can affect the induced anomalies at different levels. Additionally, the oceanic response was
found to be larger later in the season, with significant relation in both variables. This may be due to the seasonal
variability itself, as the normal climatological values for that time of the year are not seen during exceptional TC
conditions (e.g. SST is usually colder but TC prone conditions require it to be higher) (Amorim et al., 2017; Lima et
al., 2021) and the oceanic response may help the impacted area return to values closer to the climatology, in both
variables, in respect to that time of the year.
Two particular case studies were evaluated in further detail concerning hurricanes Ophelia (2017) and Nadine (2012).
Hurricane Ophelia was a particular case as it corresponds to the only major hurricane in this study region and had
almost its entire track inside this area. Ophelia showed strong induced anomalies for both Chl-a and SST variables.
Regarding Chl-a, Ophelia had a stronger impact towards the end of its track within the region, revealing that its
intensity played a key role in inducing Chl-a TC-related anomalies, with the mean modulus of wind stress revealing a
positive and significative relation to the evolution of the storm and therefore its intensity. On the other hand, Ophelia
had a stronger impact on the SST in its cyclogenesis, probably related to ocean-atmosphere thermodynamic exchanges
during its maturing. Hurricane Nadine, one of the slowest TCs in this study, showed more prominent induced
anomalies, especially regarding SST. In this case, considering the low translational speed of Nadine, the objective was
to study the impact that consecutive overlaid observations had on the induced anomalies. It is evident through this
analysis that the impact increases with the number of superposed observations, implying that Nadine's slow translation
speed and particular track geometry played a key role in creating such TC-related anomalies.
This study allowed for both the quality control of the remotely sensed "cloud-free" Chl-a and SST multi-sensor
products by comparing them to similar non-interpolated products, and in the sense that it identified expected changes
in the variables in areas covered by TC clouds and established crucial relations with some principal TC aspects. Future
studies should aim to understand the inherent physical mechanisms that affect the ocean during and after the passage
of a TC to better comprehend the associated induced anomalies.
**Code and Data availability**
All code and raw data used to support the conclusion of this article will be made available by the authors, without
undue reservation.
**Acknowledgements**
This work was funded by the Portuguese Fundação para a Ciência e a Tecnologia (FCT) I.P./MCTES through national
funds (PIDDAC) – UIDB/50019/2020. The authors also wish to thank the project "DiscoverAZORES", PTDC/CTA-
AMB/28511/2017 for all the help/collaborations. The authors would like to thank the anonymous reviewers for their
thoughtful comments, suggestions, and efforts towards improving this work.
**Author Contribution**
Miguel M. Lima: Conceptualization, methodology, software, validation, formal analysis, investigation, writing –
original draft, review and editing. Célia M. Gouveia: Validation, supervision, writing – review and editing. Ricardo
M. Trigo: Validation, supervision, writing – review and editing, funding acquisition.
**Declaration of Interests**
The authors declare that they have no known competing financial interests or personal relationships that could have
appeared to influence the work reported in this paper.

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
