# Peer review of "the Azores Region"

_EGUsphere, 2022_

## Author Response (AR1)

Initial statement / relevant changes to the manuscript:

We would like to thank the reviewers for the constructive comments on our study, they were appreciated and will certainly improve the overall quality of this article. Some points of the manuscript have suffered major revisions to answer the criticisms/suggestions made by the reviewers, including:

- Some steps of the methodology have been revised to take into account the characteristics of each Tropical Cyclone (TC), thus, we have considered the climatological situation for each individual storm and compared it to the condition when the TC occurred. This change allowed us to individually study the responses for each TC more accurately while at the same time separating the SST and Chl-a response completely. This resulted in several changes to the figures presented, namely Fig. 2 and the results from the following figures, the manuscript has been reviewed accordingly.

- The uncertainty surrounding the interpolated data was addressed in this revision. For this, we incorporated two types of analysis: 1) we showed the approximated errors associated with the analysed data and for various time periods surrounding TCs, which we include in new Fig. S1 since it does not present a substantial change to the main manuscript itself; 2) we used the previously shown two study cases (Nadine and Ophelia) as evaluation cases for non-interpolated data, which we did include as a novel Fig. 8, since this comparison is more relevant in this subject. Overall, the interpolated datasets appear to provide consistent data that delivered good results, either not showing a large uncertainty (particularly for SST) and showing good relations to non-interpolated data (particularly for Chl-a).

- Some small but important changes were made in the results section, with the addition of individual 6-hour observation analysis, which corroborated the analysis made in the original manuscript; and in the Nadine (2012) study case, which was not clear enough in the original version and is now presented in a renovated Fig. 7.

- Lastly, several changes were made across the whole manuscript document to address the smaller recommendations of the reviewers and to correct some aspects that were not yet entirely up to standard, such as the bibliography.

Overall, we are confident that these changes contributed to clarify some issues not sufficiently clear in the original manuscript. In this regard, the observations made by the reviewers were greatly appreciated and have certainly helped to improve the quality of the revised manuscript.

**Response to the 1st reviewer**

**Major comments:**

1. **As mentioned in the introduction, one important dynamical process is the Ekman pumping that sometimes leaves significant cold wake behind TCs. Is this process not important in the area of focus? The authors seem not to refer to this aspect in the results. For example, the authors suggest that thermodynamic exchanges should be important for the impact of Hurricane Ophelia on SST at the initial phase (L263-265), but Ekman pumping still remains to be examined, if I understand correctly.**

Ekman pumping is an important component of the surface mixing as shown in some of the literature we've presented in the manuscript (e.g., Prince, 1981). The Ekman pumping is often computed using satellite (wind and wind stress), however, it can be complicated to study this effect behind TCs using remote sensing data due to large gaps in daily data as a consequence of frequent cloud cover. To partially overcome this caveat, we have elected to produce an additional analysis included in the Ophelia study case that expands beyond our study region and explores the wind stress data provided by the NOAA CoastWatch dataset. This dataset is derived from wind measurements obtained from the Advanced Scatterometer (ASCAT) instrument onboard EUMETSAT's MetOp satellites (A and B). ASCAT presents a near all-weather capacity (not affected by clouds), as it operates a frequency in C-band (5.255 GHz), therefore, minimizing the number of missing values in predominately clouded areas such as the case of tropical cyclone paths.

2. **L88-90. Does the interpolation of data affect the results? Because the ocean is usually covered by clouds in TCs, the results might be affected to some extent by the interpolation. Please show what percent of the area analyzed was covered by clouds and discuss the influence on the results.**

We thank the reviewer for the very relevant point raised here. The CMEMS interpolated datasets used in this work aims to improve the low level of knowledge over those areas with strong cloud cover, however, it is expected that few data available should be affected by the interpolation. The data providers cannot guarantee absolute success in this process although with high reliability (Krasnopolsky et al., 2016 (doi:10.1155/2016/6156513); Maritorena et al., 2010 (doi:10.1016/j.rse.2010.04.002; Saulquin et al., 2019 (doi:10.1080/1755876X.2018.1552358)). CMEMS does not provide information of percent area that is covered by clouds in their data, but instead, they provide approximated information about errors (Chl-a) and uncertainty (SST) associated with the data (computed from the methods presented in Krasnopolsky et al. (2016) and Saulquin et al. (2019)). Therefore, in the revised manuscript we will incorporate this information (Figure R1) in the analysis and take it into account in the discussion of the results.

[Figure]

*Figure R1 (New Figure S1) - Value of associated uncertainty for Chl-a (top row) and SST (bottom row) for three critical moments of this analysis (before, during, and after TCs) and a random sample from the dataset. Do note the larger scale of uncertainty for chl-a.*

**3. L166-175. The method needs a bit more explanation. How does the algorithm detect the changes in SST and Chl-a? The brief summary may help understand the results. Do the authors use the same time windows for all TCs? The appropriate windows could change, depending on the properties of TCs such as the translation speed and size. Does the use of the same time windows affect the results? I expected that the decrease in SST is not always coincident with the increase in Chl-a, because their changes may depend on not only TCs but also the oceanic conditions. Can we reasonably assume that they occur at the same time? Another thing is that if we focus on the maximum anomalies during the periods after the passages, can we obtain almost the same results? I expect that the same time windows may blur the nature of the oceanic response.**

We appreciate the reviewer's comments on this matter and agree that the methodology requires further explanation and clarification. At first, the window considered was the same for all TCs, which in retrospect seems to be not the most adequate for this study. The nature of this methodology forced our algorithms to search before the storm for a mean situation and after the storm for a significant response and then produce a mean ideal window to study all considered TCs. Indeed, as it stands it is not flexible enough to accommodate for differences among these many different storms, with diverse translation speeds and sizes, as pointed out by the reviewer. We have considered to implement a major change in the methodology that is capable of better representing such differences between TCs. Thus, some steps in the methodology have changed to better account the individual characteristics of each TC. In particular, we have considered the climatological situation of that storm's time period and compared it to the observed situation when the TC occurred in the region. This new approach allows

the study of different time periods where SST and Chl-a responses differ, as well as different impacts depending on the TC's characteristics.

4. **Results. In this study, the difference in location seems to be not taken into consideration. Is it reasonable assumption that the ocean responds to TCs in the similar way in this region?**

It is true that the difference in location was not taken into consideration, and we want to thank the reviewer for this important point. In fact, as discussed in the introduction, there is a noticeable meridional gradient of each variable in this region (warmer SSTs in the south and more biological activity in the north), this matter was further explored and will be taken into consideration in the revised manuscript. The novel methodology, as described in detail when answering the 3rd and 5th major comments, allows to take this latitudinal dependence into account. Thus, we have now analyzed the response in each observation respective to the latitude and longitude of each observation (see Fig. R2), results were only significant (at the 95% statistical level) for the Chl-a respective to the latitude. However, the relation is minimal (r = 0.135), and since the other responses were not significant, we decided not to include these results in the main manuscript, but to mention them since they are relevant.

[Figure]

*Figure R2 - Relation of Chl-a (top row) and SST (bottom row) induced anomalies with latitude (left) and longitude (right). The dashed line indicates non-significance at a 95% confidence level.*

5. **L215-227. The properties of TCs change greatly while they move with time, and especially the translation speed may be quite different between the initial and mature phases, as seen from Figure 6. So, each plot in Figure 5 indeed has a broad range. I wonder if the average of the translation speed makes sense when comparing to SST and Chl-a anomalies.**

We agree with the reviewer that the properties of TCs change a great deal during their lifetimes such as seen in the Hurricane Ophelia study case. In this regard, we will have an additional change to the revised methodology, where it will be divided in full TC and individual 6-hour observations. These two approaches differ in the way they are processed since the full TC eliminates any superposition of pixels (such as seen in the Nadine study case) and allows us to analyze the area after the complete passage of the cyclone over the region; for the second, we will not have this possibility and the superposition needs to be accounted on the discussion, however, this allows the study of the responses based on the observations' characteristics (intensity, translation speed, etc.). Concluding, this change will impact the results since the revised figures will include individual observation (see Fig. R3).

[Figure]

*Figure R3 (New Figure 5) - Same as previous figure 5 (original manuscript) but taking into account individual 6-hour TC observations. An additional revision made was to incorporate the response with respect to the time in the season (d) and (h).*

Other points:

**6    L71-72. What does it mean by "in relation to the rest of the north Atlantic basin"?**

It is related to the much lower cyclonic activity observed in our study region in relation to that observed in the rest of the north Atlantic basin, in which it is inserted.

**7    Methodology. The first two paragraphs are a bit redundant, because most information is already given in the introduction and data sections. Please briefly describe the points.**

We agree with the reviewer, and we will make an effort to reduce some of these sentences in the revised methodology.

**8    Figure 7, right panels. Color panels are easy to see, like in Figure 6.**

Not entirely sure we understood this point, it is however worth saying that old figure 7 has suffered some changes to help clarify our results, as per other reviewers' suggestions.

Response to the 2nd reviewer:

Major comments:
1. **The study compared the condition of SST and chlorophyll (chl-a) before and after the passage of TCs. TC has its own lifetime which experiences increase and decrease of intensity, moving speed, impact area, and also has its own track. When you compare the passage of TC, do you use the data from all impacted area, or just individually one grid box from the TC's track? Or more specifically, in Fig. 2, does the area of "Cyclone passes over the area" represent one grid box of data, or the entire impact area of a signal TC.**

The areas studied before and after the cyclone passed over the area are the same and greatly reduced from the initial study region (red box in Fig. 1 of the original manuscript). More direct examples are the study cases seen in Figs. 6 and 7 (original manuscript), where the areas shown in subplots (a) represent the area of study for each of the cases inside the study region. This reduction cuts the heavy computational cost of analyzing the entire study region and all the TCs individual observations. For this, we used the provided approximated quadrant radius for the 34-kt isotach (the limit to be considered a tropical storm) and largely reduced the study area. This not only reduced the computational burden but also removed parts of the grid that did not have any influence from the TCs, thus improving the analysis we can provide.

2. **How is the impact area of ocean defined?**

The impact area was chosen based mainly on two factors: the lack of studies regarding the ocean response to TCs; and the fact that the TCs that usually pass through this region are much less intense in comparison to what we can normally see in the rest of the North Atlantic basin. Additionally, this is a region where TCs still maintain some tropical characteristics (warm core, symmetry, etc.) but often start suffering transition to post-tropical storms (with loss of symmetry, increased baroclinicity, overall loss of intensity). We chose these limits based on the general location of this transition, not to consider many post-tropical observations (not too much extended to the north and east) and not many purely and intense tropical storms (not too much extended to the south and west). Finally, this region is well-studied oceanography wise, with some literature exploring the main characteristics of the Azores regions in terms of SST and Chl-a (Amorim et al., 2017 (doi: 10.3389/fmars.2017.00056); Caldeira & Reis, 2017 (doi: 10.3389/fmars.2017.00037)). The exact number for the limits of the region was purely a compromise between the aforementioned points and the available data for this study.

Other points:
3   **Caption of Fig. 4, TC induced anomalies of (a) total chl-a and (b) SST.**
We will clarify.

Response to the 3rd reviewer:

Major comments:

1. **First, the authors present their use of interpolated, gap-free data as "particularly relevant in the context of this study" (l. 88), to which I strongly disagree. The presence of clouds during TCs masks the area and period of the most intense air-sea interactions, when the winds are the most intense. Interpolating from observations taken when and where there is no cloud, i.e., when and where the TC is absent, does not allow estimating the ocean state under the TC. Interpolation simply does not give access to missing observations. I suggest the authors to perform the same analysis with original, non-interpolating data. Their results should be similar, without anyone questioning the impact or role of the interpolation on their results.**

We acknowledge some of the caveats by using this interpolated dataset and the necessity to address this more carefully. The data we used was a blended product that, even though originating from several microwave sensors, will still have a large amount of error in relation to clear skies observations. Although we did not take this into account in the original version of the manuscript, we are to incorporating a performance assessment of this data in the revised version in two ways: 1) we intend to show that the approximated error associated with the data provided by CMEMS, as they do not provide any other measure of cloud cover or missing data, and, this approach aims to evaluate the overall quality of the analyzed data in respect to the equivalent non-TC area errors (see fig. R4); 2) we will use both case studies as evaluation cases for un-blended (i.e., non-interpolated) data and compare them to the original blended SST and Chl-a data from CMEMS (see fig. R5). Ideally, the second approach should be applied to all TCs, however, satellite products usually do not have such a long available time period without gaps (either due to cloud coverage, satellite blind spots, or satellite down-time), making a general study complex and requiring several different datasets to cover the entire region for over 20 years. Therefore, the datasets presented in the original manuscript were particularly relevant in this broad context. Nonetheless, we are confident that the addition of results based on the non-interpolated data will only benefit the original analysis.

[Figure]

*Figure R4 (New figure S1) - Value associated uncertainty for chl-a (top row) and sst (bottom row) for three critical moments of this analysis (before, during, and after TCs) and a random sample from the dataset. Do note the larger scale of uncertainty for chl-a.*

[Figure]

*Figure R5 (New figure 8) - Histograms for the situation before (black) and after (grey) for each of the study cases. Top row shows the CMEMS interpolated data and the bottom row the correspondent non-interpolated data.*

**2  Second, I find the presentation of the anomaly estimates confusing (l. 145 to 175 and figures 2 and 3). The authors first mention that they "analyzed daily anomalies registered between 30-days before and 30-days after the passage of a TC" (l. 149-150). With respect to what are these anomalies estimated? These anomalies are then apparently used to estimate the periods to consider before and after the TC to estimate anomalies (l. 150-155 and Figure 2). Figure 3 seems to present anomalies that are used to identify periods from which anomalies**

**are going to be estimated. Overall, it seems that there are 2 types of anomalies, but this is not clear. This whole section of the manuscript needs to be clarified**.

We agree with the reviewers' observations that there is a need to clarify how these anomalies are computed. There are indeed two types of anomalies that aim to account for two very different situations: 1) The first type is presented in the methodology section and corresponds to "classic anomalies" obtained when daily absolute values (of both SST and Chl-a) are compared to the corresponding daily climatological values for the region (Fig. 3, original manuscript); these were used to eliminate the seasonal trend in our data.  2) The second type of anomalies, shown in the results section, corresponds to "anomalies induced by the TCs", originating from the difference between the mean situation before the TC and the situation after (Fig. 2, original manuscript), only based on absolute values prior and after the passage of each TC, i.e., without using the climatological reference. Upon this and other reviewers' suggestions, the revised version of the manuscript has suffered a significant improvement on several methodological steps, including the time considered before and after the passage of TCs. In the revised approach we are only using the observed values to compute everything that is needed (search and results), i.e., no "classical anomalies" are considered. This new approach considers the climatological situation of that storm's time period and compares it to the observed situation when the TC occurred in the region, in the form of excess standard deviation and then evaluates the maximum differences before and after the storm. A practical example for Hurricane Nadine is shown for Chl-a (Figure R6).

[Figure]

*Figure R6 - Daily excess standard deviation over the climatological value of Chl-a for the area affected by Hurricane Nadine (2012).*

3   **Finally, I find that the analysis of the Hurricane Nadine case (2012) is not very convincing. Figure 7 shows very small differences in the mean SST and Chl-a anomalies estimated from the various levels of overlap. Are these differences significant? It seems very hard to conclude anything based on figure 7. I suggest**

**the authors to perform a more thorough analysis of that hurricane, or of another TC, in order to bring more convincing results to their study**.

We agree with the reviewer that the study case of Hurricane Nadine was not entirely convincing. We tried using essentially the same approach to the analysis of both case studies and presenting them as such, in retrospect this proved not to be so successful for the Nadine study case and the revised manuscript does have these observations into account (see Figure R7).

[Figure]

*Figure R7 (New figure 7) - Revised Nadine study case. Scatter plots (b) and (c) show the avg. induced response for each subregion (based on super-position of pixels) inside the affected area in (a).*

Suggestions:

    **4   L. 11, I suggest: "during and after the passage of the TC", not only after.**
Changed accordingly.

    **5   L. 22, "while dynamical mixing played a more important role in the later stage": that analysis is not found in the main manuscript.**
Since thermodynamic exchanges played a major role in the beginning of the storm, we presumed that dynamic mixing should affect more the later stages. However, without clear data to support the statement we are downgrading it in the revised manuscript.

    **6   L. 33-35, I suggest: "In his seminal study, Price (1981) shows, through both observed and numerical modelling data, the evolution of sea surface temperature (SST) etc." The grammatical structure used by the authors is not correct, or not idiomatic, in English.**
Changed accordingly

**7    L. 39-40, I suggest: "Due to the upwelling of colder water, transport of nutrient-rich water from the sub-superficial layer may also occur (Kawai & Wada, 2011)."**

Changed accordingly (original suggestion is in l. 41-42).

**8    L. 52-54: In the Bay of Bengal, surface salinity associated with river discharge is thought to play an important role.**

Yes, and this is brought in Chacko (2019), we did not include this fact to keep the paragraph concise, although the revised manuscript will include this and a better link between this paragraph and our own study region.

**9    L. 86-90 and l. 93: Please see my detailed comment about interpolated data.**

Answered in detail before (please see answer to major comment 1).

**10   L. 100, I suggest: "up to 2020", since the most recent full hurricane season is now 2021.**

Changed accordingly. At the time of writing the 2021 data was not yet available.

**11   L. 111: full tracks.**

Changed accordingly.

**12   L. 112: "the right panel showing a zoomed view…"**

Changed accordingly.

**13   L. 124: The authors can refer to Figure 1 here.**

Changed accordingly.

**14   L. 126-128: This repeats the end of the previous paragraph.**

Agree. We aim to reduce the first two paragraphs of the methodology section, also following similar suggestions of other reviewers.

**15   L. 128-129: "with fewer and less intense tropical storms".**

Changed accordingly.

**16   L. 138: "… isotach, tropical depressions were not considered (the exact partition of intensities is given etc.)"**

Changed accordingly.

**17   L. 139: "The correction of some missing…"**

Changed accordingly.

**18   L. 140: Figures 6 and 7 are mentioned before Figure 2. I suggest not to mention them here, but to mention the name of the corresponding section instead.**

Changed accordingly.

**19   L. 143: "considered area uses histograms, in which…"**

Changed accordingly.

**20 L. 149: Anomalies with respect to what? Please see my detailed comment about the anomaly estimates.**

In this instance we were referring to classical anomalies, computed from climatological values. Full answer in major comment 2.

**21 L. 158: "Schematic", not resumed schematic.**

Changed accordingly.

**22 L. 159: "The colour coding is as follows: …".**

Changed accordingly.

**23 L. 161: "Individual daily anomalies", not induced.**

In this case, it would be the newly computed induced anomalies, as they were a result of the subtraction of the daily values with the average mean before the storm. Nonetheless, this section will be changed significantly in the revised manuscript namely to accommodate the novel methodology described above when answering the major questions raised by the reviewer.

**24 L. 163: Figure 7 is mentioned before figure 3. I suggest not mentioning the figure.**

Changed accordingly. Figure not mentioned here.

**25 L. 168: Same as previously mentioned before Figure 3. I suggest not mentioning the figure.**

Changed accordingly. Figure not mentioned here.

**26 L. 171, I suggest: "period where the effects on the oceanic variables are clearly visible".**

Changed accordingly.

**27 L. 173: "too short", instead of too small.**

Changed accordingly.

**28 L. 174, "both cases": Are the authors referring to both variables?**

Yes.

**29 L. 174-175, "those extra periods varied both in number and locations": It is not clear what the authors are referring to here.**

The extra periods were relative to other time periods identified by the algorithm, and were different for SST and Chl-a, while the main ones coincided in time.

**30 L. 177: Same as previously mentioned: with respect to what are the anomalies estimated?**

Estimated with the climatological values. However, in the revised manuscript these anomalies will be disregarded.

**31  L. 184, I suggest "both variables present a large impact…"**

Changed accordingly.

**32  L. 192, I suggest "weak" instead of "weaker". Otherwise, the authors have to specify weaker that what.**

Changed accordingly.

**33  L. 199: "Fig. 4d shows", not "Fig. 4d, shows".**

Changed accordingly.

**34  L. 203: later, not latter.**

Changed accordingly.

**35  L. 204-205, "Powerful TCs induced a more varied distribution of anomalies": what exactly do the authors mean?**

We meant that the range of induced anomalies was larger. However, we realized afterwards that this statement is not correct and, therefore, will be rectified in the revised version based on new results.

**36  L. 205-207, "Do note that these different distributions do not represent the same geographical areas, since they are analyzing different observations associated with the location of each TC as it moves along its storm-track": Why are the authors mentioning this here? How should that affect the analysis of the histograms?**

It does not affect the analysis of the histogram in any way. It is just a justification for the blue and orange histograms inside each subplot (that are analyzing the same number of cyclones) not presenting the same area under the curve.

**37  L. 232, I suggest: "with a dashed line representing a regression non-significant at the 95% confidence level, and a solid line representing a regression significant at the 95% confidence level".**

Changed accordingly.

**38  L. 235: "Red squares".**

Changed accordingly.

**39  L. 238, I suggest "study region" instead of "sector".**

Changed accordingly.

**40  L. 251, I suggest "in total (grey)".**

Changed accordingly.

**41  L. 253: "maturing (Fig. 6a, squares), and mature hurricane (Fig. 6a, stars)."**

Changed accordingly.

**42  L. 254: ", and the last one, 7".**

Changed accordingly.

**43** L. 257: "therefore, the sum of all three phases will results in the bigger histogram."

Changed accordingly.

**44** L. 262-263, "with the last phase weighing the most in the general distribution (as was seen for the chl-a)": Isn't this due to the number of observations, related to the size of the storm? How does that affect the results or analysis? I am not sure this information is very relevant.

We agree, as this information is in retrospect, redundant, it will be removed.

**45** L. 263, I suggest: "The highest SST impact…"

Changed accordingly.

**46** L. 263-265: That sentence is relevant to SST only. The authors could check how robust their analysis is, maybe by looking at the SST values along the track and how they support or not strong TC air-sea interactions. Also, the authors do not analyze the Chl-a signal, which is different from the SST one. A brief analysis is included in the abstract, it should appear here in detail.

We agree with the reviewer and this issue will be one of the major points addressed in the revision in the manuscript.

**47** L. 269: The caption is in contradiction with the text. In soft grey it should be all data, not "no overlap", based on the text (l. 272-273) and on the map on Figure 7. Based on the text, in dark grey should be 3 overlaps or more, and in black 5 overlaps or more.

Indeed, we thank the reviewer for the correction.

**48** L. 274-277: As mentioned in my detailed comments, the differences seem small. Are they significant?

We did not test for significance in either of the study cases. However, this section of the manuscript has suffered a major revision and these questions will be taken into consideration for clarity.

**49** L. 277-278: That sentence is not very clear. How was the TC intensity deemed irrelevant?

We did not consider Nadine's intensity to be so relevant for this study case since the TC was not very intense and did not change too much during its lifetime in the study region. As mentioned in the previous point, this section is revised, and these points have been taken into consideration for clarity.

**50** L. 278-281: As mentioned in my detailed comments, the results derived from figure 7 are not very convincing, which makes it hard to support the hypothesis mentioned here.

We agree, and one of our aims for the revision is to make these results clearer.

**51 L. 284-286: As mentioned in my detailed comments, the use of interpolated data raises some serious questions. I suggest revisiting the analysis using non interpolated data.**

We agree with the observation (Please see answer to Major comment 1).

**52 L. 287-288, I suggest: "the existence of a bio-physical response after the passage of a TC was identified from the analysis of chl-a and SST datasets, which etc.".**

Changed accordingly.

**53 L. 288: I suggest removing the parentheses for Chl-a and SST.**

Changed accordingly.

**54 L. 289: I suggest removing "considerably", it seems a bit exaggerated.**

Changed accordingly.

**55 L. 291: "anomalies in the order of 0.026 mg/m3 in chl-a…".**

Changed accordingly.

**56 L. 293: "…oceanic responses, which agree with…"**

Changed accordingly

**57 L. 294-296: There is a contradiction in that sentence, with a relationship that was first confirmed and then not confirmed. I suggest rephrasing into something like: "In agreement with previous studies (refs), the translation speed was also found to be associated with anomalies in both variables, although the relationship was not significant at the 95% statistical confidence level in our study."**

Changed accordingly.

**58 L. 299-300: Can these various effects and their induced anomalies be documented with the database assembled by the authors?**

The two first effects were verified in our database, and the third we did not look into since post-tropical transition might occur immediately outside the study region or the best-track data does not include the observations after the tropical nature of the cyclone ceases. Nonetheless, these remarks (l. 297-300) pertained to the bibliography and not to our results.

**59 L. 309: I disagree with "It is evident through this analysis", see my previous comments.**

Agree.

**60 L. 312-313: I disagree with that sentence. The TC-induced changes should be identified on non-interpolating data. The most relevant quality control of the interpolated data would be evaluation using independent data collected under the TC, which is of course challenging.**

We agree, we detailed our new approach to this problem in the detailed comments.

---

## Author Response (AR2)

**Initial Statement / Relevant changes to the manuscript:**

We would once more like to acknowledge the effort of the reviewers in improving the quality of this work. The main points to suffer some minor revisions were:

- A paragraph in the methodology, which was addressed by both reviewers, has been clarified to better reflect the methodology used.
- Minor corrections in the text.
- Changed the unit of SST from Kelvin to degree Celsius.
- Colour schemes and symbols in figures have been updated to ensure readers with colour vision deficiencies can correctly interpret the results.

The minor revisions made according to the suggestions presented have decidedly improved the clarity and quality of this final version of the manuscript. Following, are the point-to-point responses to the reviewers.

**Report #1 – Reviewer 1**

L155-156. Does this sentence mean that missing values are estimated using a simple linear regression?
Yes, we have made the sentence clearer.

L161-165. Can you explain how the daily standard deviation is calculated. Is it the spatial RMS of the anomaly of SST and Chl-a from the climatology over the TC's area?
We understand that the daily standard deviation mentioned in the previous version of the manuscript was not clearly explained. To ensure the readability of this issue, we have changed this paragraph to make it clearer: First, we calculated the climatological mean and associated standard deviation of both Chl-a and SST values for the region that is impacted by each TC on the day of analysis. This is achieved considering the 3 days before and 3 days after the day of analysis, totalling one week that is then retrieved from the entire study period of 22 years, thus ensuring a larger sample and a smoother continuous curve. Then, we compute the mean value in the same area (in which only the TC area was considered) for the day of analysis, and finally, we calculate the normalized anomaly from the climatology on that day. Underline: For example: After a certain TC passed, we have the area with a mean SST of 300 K. The climatology for the same area is 305 K with a standard deviation of 5 K. Therefore, the mean daily value after the TC is one standard deviation below the climatology.

L379. 'it was generally identified' should be removed
Changed accordingly.

**Report #2 – Reviewer 3**

**Detailed comment**

Although I acknowledge the efforts made by the authors to improve the presentation of their methodology to estimate the TC-related Chl-a and SST anomalies, I still find this part confusing in the revised manuscript. In particular, the authors mention that they "computed for each storm the daily standard deviation of both Chl-a and SST over their respective grids relative to the climatology over the same area" (l. 161-163). What exactly is a standard deviation relative to the climatology? The associated figure, Figure 2, presents negative standard deviations, which is not mathematically correct, as standard deviations are always positive. I suspect that the authors estimated, for each pixel in the area affected by the TC, the anomaly with respect to climatology, which they averaged over that area. If that is the case, this term should not be called "standard deviation above climatology", but "anomaly with respect to the climatology". In order to avoid confusion, the term "anomaly" that is currently used in the manuscript, which represents the anomaly between before and after the TC passage, can be named "TC-related anomaly". This is just a suggestion. In any case, this part (l. 160-182) needs to be clarified.

We understand that the part of the methodology regarding the daily standard deviation was not explained sufficiently well. To make it clear, we have changed that paragraph to make it entirely clear: First, we calculated the climatological mean and associated standard deviation of both Chl-a and SST values for the region that is impacted by each TC on the day of analysis. This is achieved considering the 3 days before and 3 days after the day of analysis, totalling one week that is then retrieved from the entire study period of 22 years, thus ensuring a larger sample and a smoother continuous curve. Then, we compute the mean value in the same area (in which only the TC area was considered) for the day of analysis, and finally, we calculate the normalized anomaly from the climatology on that day. For example: After a certain TC passed, we have the area with a mean SST of 300 K. The climatology for the same area is 305 K with a standard deviation of 5 K. Therefore, the mean daily value after the TC is one standard deviation below the climatology.

Nonetheless, we made an effort to change several appearances of the word "anomaly" to TC-related anomaly or induced anomaly to make clear that we are referring to anomalies that are induced by the TC and not anomaly in the regular sense of the word.

**Suggestions**

- l. 34, "shallow layers": it should be "deeper layers".
Changed accordingly.

- l. 48: vary, not varies.
Changed accordingly.

- l. 49: "that the most impactful phenomena are intense and slow TCs".
Changed accordingly.

- l. 97-98: "Similarly to the previous CMEMS interpolated Chl-a product, the SST field etc."
Changed accordingly.

- l. 128: TCs, not TCs'.
Changed accordingly.

- l. 134: the tracks of all the TCs (not the tracks for all the TCs).
Changed accordingly.

- l. 155: "and, in order to correct for those, a simple linear regression etc."
Changed accordingly.

- l. 186-187, "(i.e., that pixel is no longer inside the radius of influence of the TC)": this is not clear, as only the pixels under the TC are supposed to be used.
This was not well explained; we meant that the observations in that pixel during the days when the TC is still over the area are discarded. We have changed the phrase accordingly.

- l. 191/Figure 2: A standard deviation cannot be negative, please see my main comment.
We have changed the paragraph that is associated with this explanation accordingly to make it clearer. Full response in the long answer to the detailed comment.

- l. 208-209: "-1.615K", rather than "negative 1.615K". I also suggest presenting SST results in deg. C.
Changed accordingly.

- l. 225: I suggest "remarkable" rather than "expressive".
Changed accordingly.

- l. 226-227: "with slower TCs having a slightly stronger impact etc."
Changed accordingly.

- l. 233-234: "zero value in a grey line".
Changed accordingly.

- l. 240: -0.3 is not a high value for a linear regression.
We agree and changed this phrase accordingly.

- l. 250: "… requires a different approach. Fig. 5 shows similar etc."
Changed accordingly.

- l. 255-256: "This caveat was not present in Fig. 4 since we considered the TC lifetime as a whole and could then disregard the days of superposition": It is not clear why the authors cannot discard the exact same days for the present analysis.
We could not discard the exact same days because the analysis is not the same for both cases. When we consider a single 6-hour observation we do not take into account

possible observations before or after, therefore analysing if that observation included pixels which had been affected becomes problematic. Additionally, superposition of pixels is not very common, as only about 10% or less of pixels have superposition from one observation to the next (see example of Ophelia's track). There are nonetheless the cases where it is problematic, like Nadine, which is the reason for the case study and why (for the general response) we removed this superposition.

- l. 261: "with respect to the time of year".
Changed accordingly.

- Figure S1: The SST is usually expressed in deg.C and not in K.
We kept Kelvin in this figure since the associated uncertainty is given for Kelvin and changing it to deg.C would require either further explanation or percentages relative to deg.C.

- l. 275-276, "as both variables were correlated with both latitude and longitude, and only negligible and non-significant relations were found". There is a contradiction here: if the relationship is not significant, there is no correlation.
We agree and have changed accordingly.

- l. 279: "uncertainty for the used Chl-a and SST datasets. Thus, we have explored etc."
Changed accordingly.

- l. 291: I suggest "likely" instead of "maybe", and "larger" instead of "higher".
Changed accordingly.

- l. 291: "cloud-covered area in that situation. After the storm etc."
Changed accordingly.

- l. 294: "Additionally, the variation that has been identified before, with Chl-a increasing and the SST decreasing, is noticeable in both variables."
Changed accordingly.

- l. 296: Remove "are marked".
Changed accordingly.

- l. 300: insert, not inset.
Changed accordingly.

- Figure S2: There is a mistake in the caption. It is Fig. 6, not Fig. 5.
Changed accordingly.

- l. 330: Fig. S2, not Fig. S3.
Changed accordingly.

- l. 332: "marker type to Fig. 6. These plots exceed etc."
Changed accordingly.

- l. 335: "storm's intensity. As Ophelia etc."
Changed accordingly.

- l. 348: 0.040 mg.m-3 is not a large anomaly, as it is lower than the mean value from Fig. 3.
We meant when comparing to the less superposed areas, we have changed accordingly to make this idea clearer.

- l. 349: It is very common in academia to use parentheses to present two opposite results in the same sentence, as is done here, but this habit is very confusing, and thus counter-productive. The reader has to read the same text twice or three times to make sure they understand it correctly. In particular, in this case, parentheses were first used earlier in the same sentence to provide additional details about the anomalies, whereas at the end they are used to mention opposite results. This is totally confusing. I can only recommend the authors to read Robock (2010, https://eos.org/opinions/parentheses-are-are-not-for-references-andclarification-saving-space). Please, join the movement!
Thank you! We have changed this phrase accordingly as it does indeed make more sense this way.

- Figure 8: Again, the SST is usually expressed in deg.C and not in K. Here it is necessary to match the analysis made by the authors, who use deg.C and not K.
We have changed thoroughly from Kelvin to deg.C in the entire manuscript, except for Fig. S1 since the associated uncertainty is given for Kelvin.

- l. 366: I suggest "does not differ much", rather than "too much".
Changed accordingly.

- l. 368: "despite both TCs presenting etc."
Changed accordingly.

- l. 372, "Nonetheless, interpolated SST data does show the less uncertainty as verified before as the process of interpolating the data fixes this issue (Fig. S1)": This sentence is not clear at all and needs to be rephrased.
The last part of the phrase was an attempt at justifying the differences seen. However, these are two different datasets therefore this attempt makes little sense. We have thus decided to remove that and rephrase the beginning of the sentence accordingly.

- l. 379: I suggest removing "it was generally identified".
Changed accordingly.

- l. 392: Larger, not increased.
Changed accordingly.

- l. 392: "… with significant relation in both variables. This may etc.".
Changed accordingly.

- l. 393-394, "this may be due to the seasonal variability of the variables themselves, as the normal climatological values for that time of the year is exceeded in exceptional TC conditions (Amorim et al., 2017; Lima et al., 2021)": This sentence is not clear at all and needs to be rephrased.

Changed accordingly for more clarity.

- l. 394-395, "the oceanic response may help the impacted area return to expected values in both variables, in respect to that time of the year": This sentence is not clear at all and needs to be rephrased. Do the authors mean: "the oceanic response may help the impacted area return to values that are closer, for both variables, to values expected from the climatology at that time of year"?

Changed accordingly for more clarity.